# Transparent multispectral photodetectors mimicking the human visual system

Qitong Li[1], Jorik van de Groep [1], Yifei Wang[1], Pieter G. Kik[1,2] & Mark L. Brongersma[1]*

Compact and lightweight photodetection elements play a critical role in the newly emerging augmented reality, wearable and sensing technologies. In these technologies, devices are preferred to be transparent to form an optical interface between a viewer and the outside world. For this reason, it is of great value to create detection platforms that are imperceptible to the human eye directly onto transparent substrates. Semiconductor nanowires (NWs) make ideal photodetectors as their optical resonances enable parsing of the multi-dimensional information carried by light. Unfortunately, these optical resonances also give rise to strong, undesired light scattering. In this work, we illustrate how a new optical resonance arising from the radiative coupling between arrayed silicon NWs can be harnessed to remove reflections from dielectric interfaces while affording spectro-polarimetric detection. The demonstrated transparent photodetector concept opens up promising platforms for transparent substrates as the base for opto-electronic devices and in situ optical measurement systems.

[1] Geballe Laboratory for Advanced Materials, Stanford University, Stanford, CA, USA. [2] CREOL, The College of Optics and Photonics, University of Central Florida, Orlando, FL, USA. *email: brongersma@stanford.edu

The newly emerging augmented reality, wearable and sensing technologies require the development of compact and lightweight photodetection elements. Such elements are used to collect optical information from the surroundings to optimize the user experience. In these technologies, the detectors should be imperceptible to the human eye and allow creation of transparent optical interfaces, e.g. between a viewer and the outside world[1]. On the other hand, the detectors should also be able to extract multi-dimensional information carried by light, including intensity, direction, wavelength, polarization state, and phase[2–9]. Specifically, spectro-polarimetric detectors are designed to collect valuable spectral and polarization information from a scene. They can be used as standalone detectors or as smart pixels in imaging systems[10,11]. Conventionally, narrowband color and polarization filters are placed in front of the detector elements to determine the spectral and polarization content. Inspired by the operation of the human visual system, algorithmic spectrometry approaches are now gaining popularity[12–15]. Here, the color-sensing elements do not have filters and instead display broad overlapping spectral responsivity curves like the cone-shaped photoreceptor cells in the retina. These cone cells support daytime vision and afford perception of color by having different cells that display sensitivity maxima at long (red), medium (green), and short (blue) wavelengths. By a learning experience, the brain can capitalize on the overlap of the spectral responsivities of the cones to distinguish subtle color differences (i.e. intensities and wavelengths). This would be impossible if the spectral responsivities were not overlapping. In that case many more sensors and filters would be required to obtain the same information. Increasingly, nanophotonic elements are used as photodetectors[3–9] or integrated with image sensors[16–23] to tailor their spectral response. Given the current desire to place detectors on transparent optical elements without being seen, we have the additional challenge that they should not reflect light or perceivably distort optical wavefronts.

High refractive-index semiconductor nanowires (NWs) are ideally suited to realize miniature spectro-polarimeters. They afford convenient charge extraction and support optical Mie resonances[24,25] that can be used to tailor their spectral and polarization-dependent absorption[5–9,18]. Based on this knowledge, it is possible to realize color and polarization sensitive detection pixels with overlapping responsivities in the red, green, and blue (see Fig. 1a). Invisible, transparent conductive contacts are well-established and can be used to extract the generated photocurrent from the NW terminations. However, it is non-trivial to make the detectors themselves undetectable. We envision the creation of detector pixels capable of tapping off a small fraction of the incident light to extract information about the optical environment, while being imperceptible to the human eye. In an ideal case, the placement of such detectors on a dielectric substrate would remove the light reflection from the surface and instead use this light to create useful photocurrent. One challenge is that Mie resonant NWs not only strongly absorb light but also exhibit strong backscattering[2,26,27]. A second design challenge is that high-index nanostructures typically feature multiple resonances across the visible range. Combined, these requirements pose stringent conditions on the NW geometry and size.

In this work, we realize imperceptible detector pixels by engineering silicon (Si) NW arrays supporting degenerate optical resonances that display an opposite (even and odd) field symmetry with respect to the plane of the array. The scattered-fields produced upon excitation of these resonances can be made to interfere destructively in the backward direction and constructively in the forward direction. In this configuration, the backscattering is thus suppressed in a broad wavelength range to achieve an antireflection function while absorption is enhanced on resonance. Moreover, forcing different resonances to be degenerate facilitates color detection with a single-peaked absorption spectrum. This contrasts the achievement of directional scattering by high-index nanostructures through an alignment of the electrical and magnetic moments at the renowned Kerker condition, which is achieved at one well-defined wavelength[28–33]. Degenerate optical resonances have been realized in nanoscale Si disks by spectrally overlapping electric and magnetic resonant modes through a judicious choice of their dimensions[34]. Unfortunately, such degenerate optical resonances cannot be found for single NWs, as we show below. Here, we demonstrate how we can nonetheless add one degree of freedom to the system by engineering not only the optical resonances of individual NWs but also the radiative coupling between adjacent NWs to achieve the desired degeneracy for NW arrays.

## Results

**Description of the transparent resonant NW arrays.** When designing detectors for the visible spectral range, it is important to realize that the lowest-lying resonance frequencies for NWs are observed for transverse magnetic (TM) polarization with the electric field along the NW axis. As a result, the optical response for small-diameter Si NWs (e.g. 10–100 nm) is dominated by the TM response and the transverse electric (TE) resonances are conveniently shifted towards the ultraviolet. Figure 1b illustrates this point for an array of rectangular cross-sectional Si NWs that have a width $w = 55$ nm, a height $h = 110$ nm, and are spaced at a period $p = 280$ nm to achieve strong absorption in the green. Based on the dominance of the TM response, we first optimized the red, green, and blue detector arrays for this polarization state. Figure 1c shows that arrays with nicely overlapping, but shifted absorption spectra can be realized.

In order to understand the symmetry properties of the resonances supported by rectangular NWs, it is helpful to view them as Fabry–Pérot-style resonators in which light can oscillate in either the horizontal or vertical direction[35]. An approximate resonance condition for the different-order resonances can be written as

$$n_{Si}^2 (2\pi/\lambda)^2 = (m\pi/w)^2 + (n\pi/h)^2, \qquad (1)$$

where $n_{Si}$ is the refractive index of Si and $m$, $n$ count the number of antinodes in the dominant field component along the horizontal and vertical directions. We find that the first-order TM resonance exhibits a symmetric field profile in the vertical direction, while the second-order resonance is anti-symmetric with two antinodes, as shown in the insets to Fig. 1h. From Eq. (1) it is clear that for single NWs the optical resonances with opposite symmetry are naturally occurring at different frequencies (blue curve in Fig. 1h). In contrast to single nanoparticles, single NWs intrinsically do not support degenerate even and odd resonances for any cross-sectional geometry (Supplementary Fig. 1). The directional scattering recently observed in a single NW[33] is the result of the Kerker condition with two spectrally misaligned resonances. Therefore, single NWs show strong backscattering at other wavelengths and cannot be used for color detection (no single-peaked absorption spectrum).

Next, we illustrate how degenerate even and odd resonances can be engineered in a NW array by tailoring the radiative coupling between them. To highlight the basic physics, we first analyze a single Si NW in air and assume a constant index of 4 for the Si material. Its first-order resonant mode features an even symmetry. It has an omnidirectional radiation pattern and a very low quality factor ($Q \sim 1$) that results from efficient far-field radiation. In an array of these Si NWs, the in-plane radiation gives rise to the formation of a new resonant mode with a

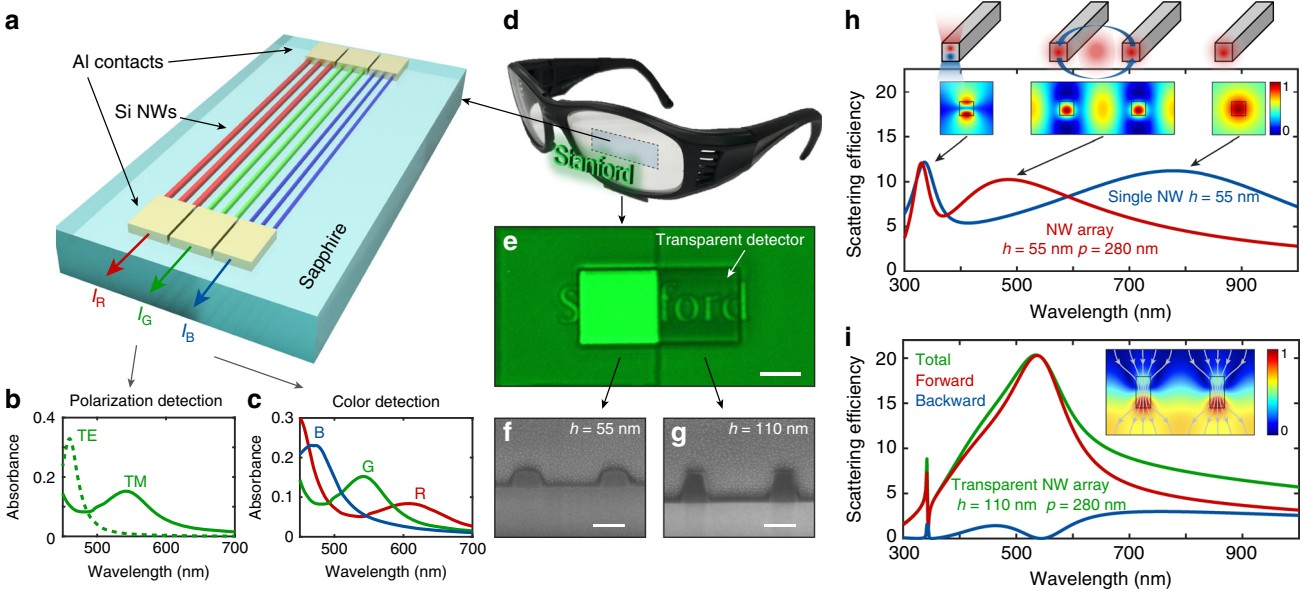

**Fig. 1** Transparent multispectral photodetector concept. **a** A conceptual design of a transparent multispectral photodetector composed of three different single-crystalline Si nanowire (NW) arrays on a sapphire substrate designed to strongly absorb red, green, or blue light. Photo-excited charge carriers are extracted via aluminum contacts. **b** Simulated absorbance spectra for the "green" NW array ($w = 55$ nm, $h = 110$ nm, $p = 280$ nm) for TM and TE-polarized illumination. **c** Simulated absorbance spectra for TM-illumination of NW arrays optimized for red, green, and blue light detection. **d** Image illustrating the possible application of our proposed detectors onto an optical element, such as a pair of glasses. **e** Reflection images of the word "Stanford" placed below square regions with conventional, reflective (left), and transparent (right) NW arrays optimized for operation in the green. The images were taken using white, TM-polarized illumination and collected through a green color filter with a center wavelength of 550 nm and a 32 nm bandwidth. Scale bar: 25 μm. **f**, **g** Cross-sectional scanning electron microscope (SEM) image of conventional (**f**) and transparent (**g**) NW arrays that are optimized for operation in the green. Scale bar: 100 nm. **h** Scattering efficiency spectrum for a single square cross-sectional NW (blue, $h = w = 55$ nm) and its array (red, period $p = 280$ nm) under TM-polarized illumination. The efficiency is normalized by the physical cross-section of the NWs. Insets: Modulus of scattering electric field distribution ($|E_{sca,z}|$) at the first-order optical resonance for a single NW (right) and NW array (center), and at the second-order resonance for a single NW (left). **i** Forward, backward, and total scattering efficiency spectra for a transparent NW array ($w = 55$ nm, $h = 110$ nm, $p = 280$ nm). Inset: Map of the scattering electric field distribution ($|E_{sca,z}|$) at the degenerate optical resonance wavelength. The gray lines show the power flow through the Si NWs as calculated by optical simulations

scattering spectrum that is notably blueshifted from the original, single NW resonance. The solid red line in Fig. 1h shows the scattering spectrum for an array with a 280 nm period. An analysis of the field distribution at $\lambda = 500$ nm, near the resonance peak, indicates that this can be viewed as a different type of Fabry–Pérot-style resonance where light oscillates back and forth between neighboring NWs. As a result, the associated resonance frequency is controlled by the NW spacing (inset to Fig. 1h and Supplementary Fig. 2). This resonant mode maintains the even symmetry seen for the single NW resonance, but features a higher Q. The second-order mode of a single NW features the desired odd symmetry that is needed to cancel backreflections. Due to its higher quality factor ($Q\sim10$) and out-of-plane, vertical radiation pattern, the coupling between NWs is negligible for this resonance. As a result, its spectral location remains the same in the array (see Fig. 1h). The first-order (even) and second-order (odd) resonances can thus be shifted independently, enabling the creation of NW arrays with degenerate optical resonances at any wavelength in the visible range.

Based on this insight, we first tune the even and odd resonances to a target wavelength of 550 nm and fabricate the resulting geometry in an Si film on a transparent sapphire substrate by electron-beam lithography. Simulations show a single, broad scattering resonance that is peaked around 550 nm (green line in Fig. 1i), attributed to the two degenerate optical resonances. The scattering from the NW array shows a giant asymmetric distribution with almost no backscattering (red, blue lines in Fig. 1i), which is also corroborated by the scattering field

distributions (inset to Fig. 1i). For this subwavelength NW array, there are no allowed diffracted orders and the scattered light is forced into the forward direction. Using a multipole decomposition of the simulated full-fields of the NW array[36–38], we verify the mode degeneracy (Supplementary Fig. 3). A similar NW array with a non-optimized, near-unity aspect ratio (Fig. 1f) is also fabricated next to the optimized array (Fig. 1g) for comparison. Figure 1e shows the reflection optical image of the NW arrays with the word "Stanford" written underneath. The reflected light was filtered for the targeted resonance wavelength and polarization (TM polarization, center wavelength 550 nm, FWHM = 32 nm) to highlight the feature induced by degenerate resonances. For the engineered NW array, light is first transmitted through the NW array, reflected by the word "Stanford", and finally transmitted back through the NW array. The clear visibility of the logo confirms the transparency induced by the degenerate resonances. On the other hand, for the non-optimized NW array with non-degenerate resonances, strong backscattering from the NWs prevents us from seeing anything behind the array. A more detailed comparison between the "conventional" and "transparent" NW arrays is summarized in Supplementary Fig. 4.

**Experimental demonstration of transparent detection pixels.** Using the approach described above, we also design and fabricate transparent NW arrays with degenerate optical resonances at the wavelengths of 485 nm (blue) and 625 nm (red), as described in Supplementary Fig. 5. Unlike degenerate optical resonances in

nanodisks—which require a fixed aspect ratio and thus different heights for different resonance wavelengths—NW arrays can be made transparent at different wavelengths in the visible using a fixed height of 110 nm. This facilitates multiplexing of different types of transparent pixels in a single patterning step. The simulated reflection of these designed NW arrays is less than 1% at the designed wavelengths for TM-polarized illumination (Supplementary Fig. 6). We measure the reflection (Fig. 2a) and transmission (Fig. 2b) spectra for TM polarization of the three different NW arrays in a confocal optical microscope and achieve good agreement with the simulated results. For example, the minimum reflection observed is 3.5% around 625 nm for the "red" pixel, clearly lower than the reflection from the bare sapphire substrate, and with a transmission/reflection ratio of 25. The small wiggle observed around 600 nm in the transmission spectrum for "red" pixel is due to the excitation of a guided mode resonance, which can be excited at off-normal incidence or with the gently focused light used here[39,40] (Supplementary Fig. 7).

To assess the performance of this detection platform for randomly polarized light, it is also important to analyze the case of TE-polarized illumination. Interestingly, the anisotropic NW array can function as a good antireflection (AR) coating for both polarizations. For TE polarization, all optical resonances exist in the ultraviolet spectral range, except the first-order resonance for "red" pixels (Supplementary Figs. 8 and 9). This resonance results in a non-perfect AR behavior, but does not prevent the use of the NW arrays for spectro-polarimetry. For the cases that the NW array is off-resonance, its effective index can simply be estimated using

first-order effective medium theory ($\varepsilon_{eff} = \varepsilon_{air}\varepsilon_{Si}/(f_{air}\varepsilon_{Si} + f_{Si}\varepsilon_{air})$). The resulting effective refractive index ($n_{eff,b} = 1.07$, $n_{eff,g} = 1.11$, $n_{eff,r} = 1.22$) is near the geometric mean of the indices of air and sapphire ($n_g = 1.33$), affording good antireflection behavior. The AR behavior for both polarizations can be seen from an analysis of the reflection and transmission spectra as well as the reflection and transmission optical images of two "green" arrays with orthogonal NW orientations (Fig. 2a, b and insets). The optical images were taken with linearly polarized light through a broadband green color filter (center wavelength = 550 nm, FWHM = 32 nm). These results indicate that the proposed NW array could be used as a transparent spectrally selective and polarization-sensitive detector.

Figure 2c shows the averaged transmission spectrum measured from 25-μm-sized "red", "green", and "blue" pixels (top insets in Fig. 2c). Transmission optical images of two multiplexed NW arrays with pixel sizes of 5 μm (bottom left) and 2 μm (bottom right) are also shown in Fig. 2c. The consistency of the pixel color at different multiplexing dimensions indicates that the NW array detector can be scaled down to sub-10 μm, comparable to color filter arrays used in commercial digital cameras. Keeping in mind that the typical transmittance for sun glasses is around 20% as a reference, the observed white-balanced transmission spectrum with a designed ~70% overall transmittance for both polarizations opens up the opportunity for in situ detection process under daylight illumination. If a lower transmittance is acceptable, NW arrays that offer an increased photocurrent can be realized. In order to visualize the transparency of the multiplexed NW arrays,

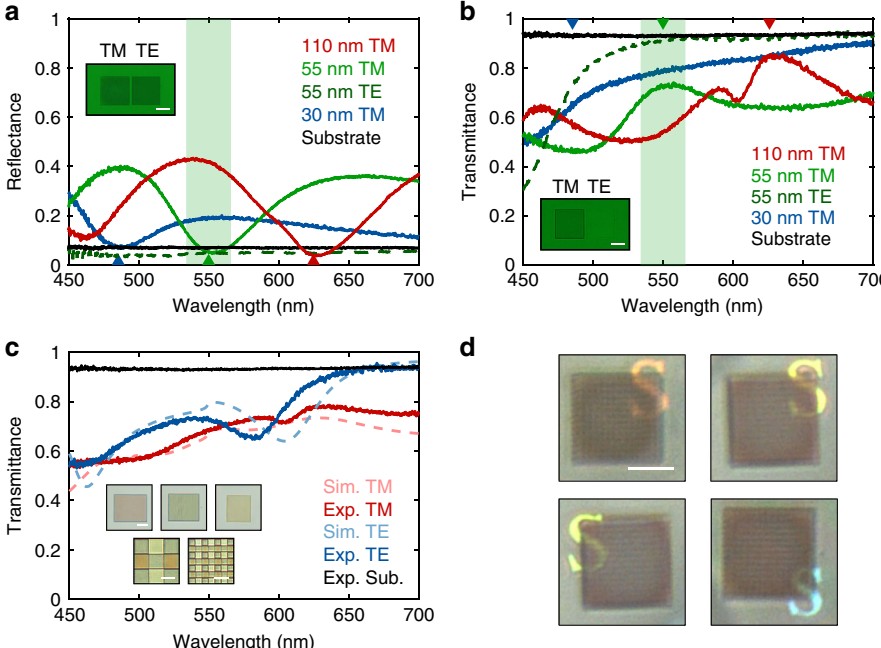

**Fig. 2** Transparency of fabricated NW array pixels. **a** Reflectance and **b** transmittance as a function of wavelength for three different NW array pixels designed for red ($w = 110$ nm, $p = 310$ nm), green ($w = 55$ nm, $p = 280$ nm), and blue ($w = 30$ nm, $p = 230$ nm) light for both polarizations. The inverted triangles indicate the wavelengths at which the NW pixels show zero reflectance in the simulations. The reflectance and transmittance of the substrate (black) is also included for reference and the reflection from the back surface of the sapphire substrate is removed in post data-processing. Insets: **a** Reflection and **b** transmission optical images for "green" pixels for two orthogonal polarizations. The incident light is polarized in the vertical direction and the NWs are aligned in the vertical direction for the left pixel and in the horizontal direction for the right pixel, respectively. A green color filter was used. Scale bar: 25 μm. **c** Overall white-balanced transmittance of three different NW array pixels as a function of wavelength for both polarizations. Insets: Transmission optical images of 25 μm size (top), 5 μm size (bottom left), and 2 μm size (bottom right) NW array pixels under TM-polarized illumination. Scale bar: 10 μm (top) and 5 μm (bottom). **d** Optical reflection images of multiplexed 5-μm-size NW array pixels under unpolarized white-light top illumination. Different colored letters "S" are placed under the pixel array. Scale bar: 25 μm

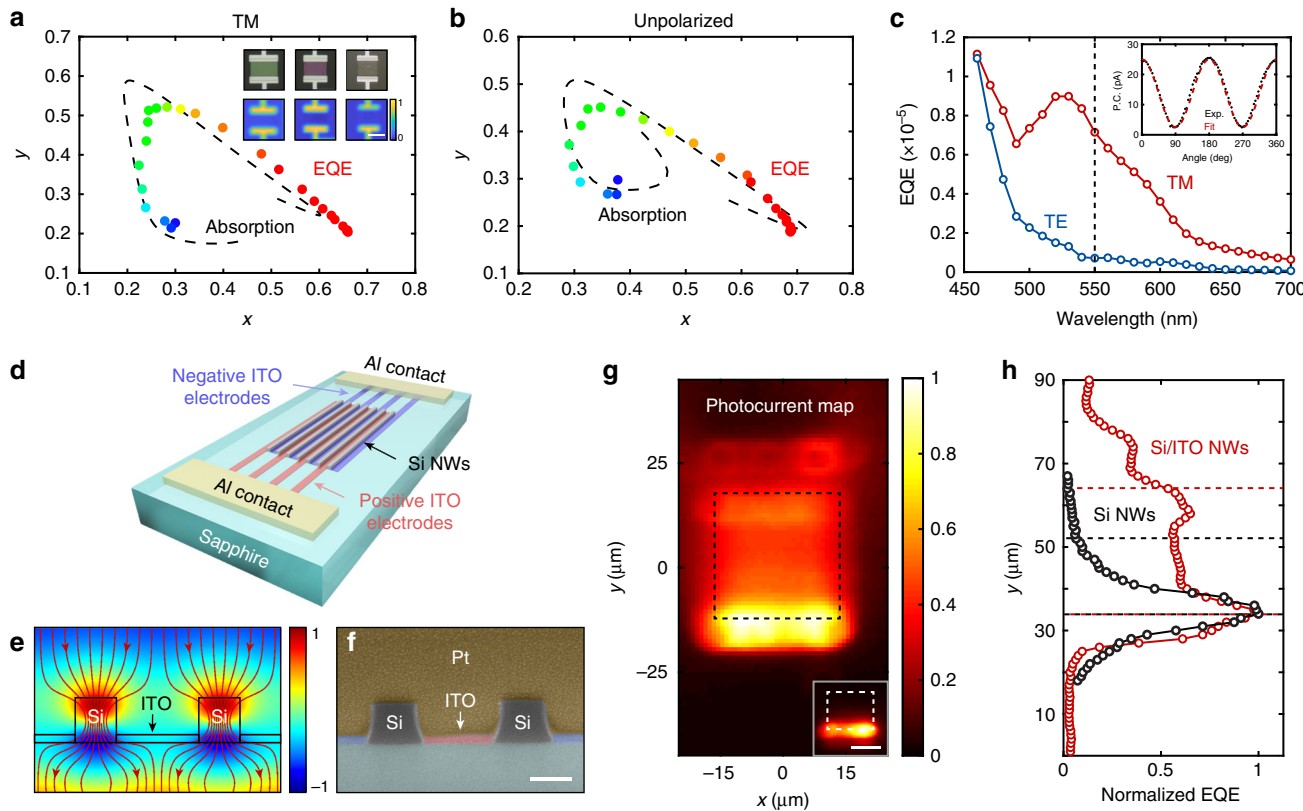

**Fig. 3** Color and polarization detection and uniform photocurrent generation. **a** The color space defined by the extracted photocurrent value from three fabricated NW array photodetectors for TM-polarized and **b** for unpolarized illumination. The dashed back line shows the simulated absorption ratio as a function of wavelength between 450 and 700 nm. The colored dots show the measured EQE ratio from 460 to 700 nm in steps of 10 nm. The x-axis is defined as $I_R/(I_R + I_G + I_B)$ and y-axis is defined as $I_G/(I_R + I_G + I_B)$. For monochromatic illumination, the photocurrent ratio is equivalent to the absorption ratio and the EQE ratio. Insets: Reflection optical images (top) and reflection maps (bottom) of three NW photodetectors. The reflection maps are taken at 625, 550, and 485 nm, respectively. Scale bar: 20 µm. **c** Measured "green" pixel EQE as a function of wavelength for two orthogonal polarizations. Inset: Measured photocurrent at 550 nm as a function of the angle between the incident polarization and the NW orientation. **d** Schematic of a Si/ITO interdigitated NW array photodetector. Photocurrent is extracted transversely across the Si NWs. **e** Electric field distribution and the power flow lines of the proposed Si/ITO interdigitated photodetectors under normal incidence. **f** False-color cross-section SEM image of the as-fabricated interdigitated Si/ ITO NW array photodetector. Scale bar: 100 nm. **g** Normalized photocurrent map of the interdigitated NW array photodetector (inset: photocurrent map for bare Si NW array photodetector. Scale bar: 20 µm) at a wavelength of 625 nm for TM polarization. The black (white) dashed box shows the position of the Si/ITO interdigitated (Si) NW array detector. **h** Normalized EQE of Si/ITO interdigitated NW array detector (red) and bare Si NW array detector (black) as a function of the position along the NWs for x = 0. Two dashed red (black) lines indicate the Si/ITO interdigitated NW array detector (Si NW array detector) area

we place objects with different colors under the multiplexed NW pixel array and take reflection optical images under unpolarized white-light illumination (Fig. 2d). All the objects with colors from blue to red can be clearly recognized, verifying the good transparency for practical applications.

**Color and polarization detection.** Next, we apply two aluminum contacts at the ends of the NW arrays to form transparent Al–Si–Al photodetectors. Besides detection of the light intensity, our transparent photodetectors also extract color and linear polarization information of the incident light by the carefully tuned resonant nature of the Si NWs. We apply a small external voltage (1 V) to drift the photo-generated free carriers to the metal contact pads and collect them. Assuming the internal quantum efficiency (IQE) is wavelength-independent in the visible range, the external quantum efficiency (EQE) becomes linearly dependent on the absorption efficiency $\eta_{abs}$ of the array (EQE = IQE × $\eta_{abs}$). We verify that the spectral dependence of the measured EQE spectra taken from the three NW array detectors

agrees well with the simulated absorption spectra (Supplementary Fig. 9). Due to the resonantly enhanced absorption, the designed NW arrays absorb red, green, and blue light separately. This affords color detection analogous to conventional Si diodes with RGB color filters, or the three types of light-sensitive cone cells in human eyes. Unlike our ultrathin transparent detector, traditional photodiodes with color filter arrays and cone cells are micrometer thick and non-transparent. Analogous to the well-established 1931 Commission International de Eclairage (CIE) chromaticity diagram, we can define a color space using the extracted photo-current value from three photodetectors as tristimulus values[41,42]. Figure 3a shows the outline of this color space calculated from the measured EQE and simulated absorption ratio of three transparent photodetectors under TM-polarized monochromatic illumination between 460 nm and 700 nm. The measured EQE ratio agrees very well with the simulated absorption ratio and the outline shape of the generated color space traces a wide trajectory in the CIE diagram (Supplementary Fig. 10), justifying the color detection ability. Since TM polarization dominates the optical

absorption, the color detection also works well for unpolarized incident light (Fig. 3b), which is essential for many practical applications. Moreover, for each color pixel, the strong polarization-dependent Mie resonance naturally enables polarization detection within each color band, as shown in Fig. 3c. The photocurrent ratio between two adjoined photodetectors with orthogonal NW orientation is dictated by the polarization of the incident light.

Some of the known challenges with NW-array detectors are related to increased carrier recombination and a reduced charge extraction efficiency as compared to bulk detectors[43]. Also in our devices, the highest photocurrents are produced near the electrical contacts (inset in Fig. 3g). This is primarily caused by the short and different diffusion lengths for electrons and holes in these etched Si NW arrays. Here, we show how we can maintain a high transparency of our detectors while promoting a more uniform photocurrent generation by applying indium-tin-oxide (ITO) interdigitated electrodes, as illustrated in Fig. 3d. Two sets of transparent electrodes with opposite DC polarity are interdigitated between the Si NWs. The 20-nm-thick ITO electrodes introduce negligible changes to the optical properties of the Si NW array (transparency and strong absorption per unit volume), because most of the power flows through the Si NWs without the interaction with ITO electrodes (Fig. 3e, Supplementary Figs. 11–13). At the same time, the photo-excited charge carriers are extracted transversely across the NWs such that free-carrier diffusion distance decreases notably from tens of micrometers to tens of nanometers. Figure 3f shows a cross-sectional SEM image of the as-fabricated device. Robust contact areas between ITO electrodes and Si NWs can be observed, with exactly the same geometry as illustrated in Fig. 3e. A spatially resolved photocurrent map at resonance is shown in Fig. 3g. The EQE signal uniformity is greatly enhanced with only 15% signal fluctuation over 80% of the device area (red lines in Fig. 3h). Unlike the exponentially decaying photocurrent generation along the bare Si NWs (black lines in Fig. 3h), the Si/ITO interdigitated detector displays quasi-uniform photocurrent generation area in the middle part of the detector. It enables a reliable intensity detection process for practical applications.

## Discussion

It should be noted that, in this study we have not aimed to optimize the electrical contacts or the NW surface passivation and the measured EQE is on the order of $10^{-5}$ (Fig. 3b). Despite the low EQE, we emphasize that the spectral shape of the EQE spectrum and the uniform photocurrent generation provide an irrevocable proof of the proposed concept. These measurements demonstrate reliable color, polarization, and intensity detection in a transparent ultrathin photodetector geometry. Mature processing techniques, including standard passivation techniques, a p–n junction, and local heavy doping at the contacts can improve the EQE of NW photodetectors for practical applications. The remaining photocurrent peak near the contact may be due to the relatively high resistance of ITO electrodes, and thus can be eliminated by optimized ITO deposition and annealing conditions. Additionally, the average transmittance can be improved further by multiplexing two different sized NWs together. This will broaden the effective bandwidth of the second-order optical resonance, leading to a lower average reflection (Supplementary Fig. 14).

In summary, we demonstrate how ultrathin NW photodetectors can be integrated on top of transparent optical elements while allowing them to function as an optical window. These proposed detection elements are realized in a single materials system (Si) and offer a high transparency, a low reflectance as well

as spectro-polarimetric functions that typically require the use of several bulky, discrete components. The demonstrated color, polarization, and intensity detection abilities can find use in a multitude of applications that range from image sensing to optical communications. The transparent multispectral photodetector concept shows the significant promise for transparent substrates as the base for optoelectronic devices and in situ optical measurement systems.

## Methods

**Numerical simulations.** We perform the 2D simulations in the frequency domain using the commercial software package COMSOL. In eigen mode simulations (Supplementary Fig. 1), the simulation area is a circle with a radius of 500 nm surrounded by a 250-nm-thick perfect matching layer (PML). The scattering field simulations (Fig. 1h, i, Supplementary Fig. 2) of single NW are performed in a 1 μm box with 200-nm-thick PML surrounding it. PML conditions are replaced by periodic conditions and scattering boundary conditions (SBC) for the NW array. To simplify the analysis of the resonant modes, Si NWs are all suspended in the air with a constant refractive index of 4 in eigen mode and scattering efficiency simulations. In the full-field simulations (Figs. 1b, c and 3a, b, Supplementary Figs. 4, 6–9, 11, 13, 14), we characterize the reflection, transmission, and absorption of NW array with a normally incident plane wave. We use periodic boundary conditions and two ports along the light propagation direction to simulate the properties of the arrays. The refractive index of sapphire is set as 1.77 and the real dispersion and loss of Si are included in all full-field simulations.

**Sample fabrication.** (1) Si NW array: We start the fabrication with a 1 cm square 500-nm-thick single-crystalline Si on sapphire piece (MTI-Corp). Reactive-ion etching is used to thin down the Si film to 110 nm (±3 nm). Seventy nanometer hydrogen silsesquioxane (HSQ) is then spin coated on the Si film (4% HSQ, 4000 r. p.m. for 40 s) to serve as negative tone electron-beam resist layer. After a 45 min baking at 90 °C, a thin conductive polymer layer (E-Spacer 300Z) is then spin coated on the Si film to reduce the charging effect during the electron-beam lithography process (JEOL 6300 100 kV system). The typical e-beam dose is set to be ~2000 μC cm$^{-2}$ and the development is performed in 25% tetra-methylammonium hydroxide (TMAH) for 2 min. Reactive-ion etching is used again to transfer the HSQ hard mask patterns to the silicon slab and the remaining HSQ hard mask pattern is removed in diluted 2% hydrogen fluoride (HF) solution for 1 min. (2) Si/ITO interdigitated NW array: We start the fabrication with 500-nm-thick single-crystalline Si on sapphire piece (MTI-Corp). Reactive-ion etching is applied to thin down the Si film to 110 nm (±3 nm). The sample is pre-baked for 60 s at 180 °C and then a 120 nm CASR layer, serving as positive-tone electron-beam resist, is spin coated (4% CSAR, 2000 r.p.m. for 40 s) on the Si slab. The sample is then post-baked for 60 s at 180 °C and E-spacer is also spin coated to reduce charging effects. We conduct electron-beam lithography to expose the ITO electrode area with a dose ~250 μC cm$^{-2}$. The development is performed in Xylenes for 40 s and reactive-ion etching is then applied again to form grooves for ITO electrodes. We then deposit a 20-nm-thick ITO film in the Si grooves by sputtering (Kurt J. Lesker Sputter). Finally, both the ITO and CSAR remaining on the top of Si NWs are removed in Remover PG at 60 °C for 48 h and the whole sample is annealed at 350 °C for 10 min to increase the conductivity of ITO electrodes. The above fabrication procedure is summarized in Supplementary Fig. 15. (3) Al contacts: Hexamethyldisilazane (HMDS) is first coated on the sample as a primer for photoresist. ~1.5-μm-thick positive-tone photoresist (MEGAPOSIT SPR 3612) is then spin-coated (5500 r.p.m. for 40 s), followed by a 60 s pre-bake at 90 °C. Optical lithography is conducted using Heidelberg MLA 150 and 60 s post-bake at 110 °C is performed before the development in MEGAPOSIT MF-26A for 40 s, followed by another 60 s bake at 110 °C. 100-nm-thick Al is deposited later in the Kurt J. Lesker electron-beam evaporator. Finally, the sample is immersed in acetone for 3 h to complete the lift-off process.

**Optical and photocurrent measurement.** (1) Bright-field reflection and transmission measurements: We perform the optical spectrum measurement using a Nikon C1 confocal microscope. Light from a halogen lamp is top (bottom)-illuminated on the sample through a ×20 objective (NA = 0.4) (condenser lens, NA = 0.4) for reflection (transmission) measurement. The reflection (transmission) signal is then collected by a ×20 objective (NA = 0.4) and polarization-filtered by a rotatable polarizer for polarization-resolved measurement. Finally, a confocal scanner with a 60 μm pin hole is used to spatially select the signal which is analyzed using a SpectraPro 2300i spectrometer (150 lines/mm, blazed at $\lambda = 500$ nm) and Pixis Si CCD (−70 °C detector temperature). The reported spectra are the average of 20 frames (0.5 s integration time each). All the reflection spectra are normalized by the reflection spectra of a protected silver mirror (Thorlabs, PF10-03-P01) and all the transmission spectra are normalized by the polarized transmission spectrum of sapphire substrate to eliminate the anisotropic optical properties of sapphire substrate. (2) Photocurrent measurements: Photocurrent measurements are performed using a home-built optoelectronic setup. A supercontinuum laser and acousto-optic tunable filter (both Fianium) are used to tune the wavelength of the

monochromatic illumination (~5 nm bandwidth). A mechanical chopper wheel (400 Hz) is used to modulate the laser light. A broadband polarizer is used to control the input polarization and a ×50 long working distance (Mitutoyo M Plan APO NIR, NA = 0.42, 20 mm working distance) objective focuses the light onto the sample. An imaging system with two 50:50 beam splitters (on flip mounts), a halogen lamp with diffuser, and CCD imaging camera with tube lens are used to image the sample. A glass slide directs a small fraction of the reflected laser light onto a large-area Si photodiode (New Focus, model 2031), connected to a lock-in amplifier (Stanford Research SR810 DSP), to measure the reflection signal. The sample is mounted on a three-axis piezo stage to accurately control the spatial position of the focused laser on the sample. A sourcemeter (Keithley 2612) is connected in series with a tunable current-to-voltage amplifier and the wire-bonded sample and applies a DC bias (1 V) to extract the generated charge carriers. The modulated amplifier output voltage is sent to a second lock-in amplifier (Stanford Research SR810 DSP) to measure the photocurrent. To calculate the EQE, the power spectrum at the position of the sample is measured using a calibrated power meter (Thorlabs, PM-100USB). The above experimental setup is described graphically in Supplementary Fig. 16.

## Data Availability
All data are available from the authors on reasonable request.

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

## Acknowledgements
This work was supported by a single investigator award from the AFOSR (Grant FA9550-17-1-0331) and a Multi University Research Initiative (MURIs FA9550-14-1-0389). J.v.d.G. was also supported by a Rubicon Fellowship from the "Nederlandse organisatie voor Wetenschappelijk Onderzoek (NWO)". We gratefully acknowledge Ankun Yang and Matthew Morea for their help with the ITO deposition. Part of this work was performed at the Stanford Nano Shared Facilities (SNSF), supported by the National Science Foundation under award ECCS-1542152.

## Author contributions
Q.L. and M.L.B. conceived the idea for this research. Samples were fabricated by Q.L., J.v.d.G., and Y.W. Q.L. performed the simulations and optical measurements with the help of J.v.d.G. All authors were involved in analyzing the data and writing the manuscript.

## Competing interests
The authors declare no competing interests.
