## [Peer Review File · Nature Communications]

Reviewers' Comments:

Reviewer #1:

Remarks to the Author:

Dear Editor,

In this manuscript, the authors propose and demonstrate a concept of transparent photodetectors with spectral and polarization sensitivity. The concept is based on resonant backscattering/reflection cancellation in silicon nanowire arrays arising from resonant mode interference inside the arrays. Though the concept of reflection cancellation based on electric and magnetic dipole resonances in dielectric nanodisks has already been studied for the design of Huygens' metasurfaces, the authors show that this concept works in nanowire arrays differently, necessarily requiring array modes to cancel the back scattering. Also, they show a clear application path for such transparent arrays to design invisible, color and polarization sensitive photodetectors and analyze many parameters required for this application (photocurrent generation, CIE diagram, etc.). So, overall I think that this is a good work, which deserves publication in Nature Communications. Here are a few additional comments, which the authors can take into account to further improve their manuscript:

1) Analysis in Fig.1i would benefit from plotting both forward and backward scattering efficiencies for the nanowires. Also, the description states that this is the scattering from the nanowire array (rather than single nanowires). Could the authors specify how is the scattering from the arrays defined? I would expect the infinite arrays to be described in terms of transmission/reflection and only single nanowires or finite structures in terms of scattering.

2) The authors write that "single NWs intrinsically do not support degenerate even and odd resonances for any cross-sectional geometry (Supplementary Fig. 1)". This is quite important statement that motivates the authors to use the coupling between the nanowires in the arrays as an additional degree of freedom. However, the analysis shown in Supplementary Fig. 1 is not so straightforward to understand. To make it more clear I suggest the authors to plot spectral positions of different modes of a single nanowire for different aspect ratios (both below and above unity, at the moment only >1 aspect ratios are considered). This would simplify this analysis and clearly show whether or not the relevant modes can be spectrally overlapped.

3) The microscope images of the arrays shown in the insets in Fig.2a&b do not seem to be consistent with the spectra. The array is completely not seen in the transmission image for TE polarization, while the actual transmission is far from unity. At the same time, the same array is seen in the reflection image, while its actual reflection is very close to that of the substrate.

4) In the caption to Supplementary Fig.7, first (d) should be changed to (b).

Reviewer #2:

Remarks to the Author:

Li et. al. have proposed a photodetector composed of an array of silicon nanowires which is transparent and color sensitive. The interplay between the two degenerate optical modes with even and odd radiation patterns results in a destructive interference for the reflection but constructive interference for the transmitted light. This leads to an enhanced transparency while the photodetector can absorb a fraction of light at one of the optical modes of the nanowires. By changing the nanowire width and the spacing between the nanowires, the authors have shown that it is possible to control the absorption spectral location and hence by connecting them in parallel, a color sensitive with high transparency and high white balance is achieved. They have also used interdigitated contacts to improve the photocurrent uniformity.

Overall, the paper is written well, and the results are almost convincing. However, because of the following reasons I think the novelty of the work is average:

- 1) The main claim of the paper is the suppression of the reflection using silicon nanowires array. The reflection of a silicon nanowire array has thoroughly been studied previously (e.g. Connie J. Chang-Hasnain and Weijian Yang, "High-contrast gratings for integrated optoelectronics," *Adv. Opt. Photon.* 4, 379-440 (2012)). Although the previous works have mostly focused on achieving strong reflection using high contrast grating with period around the wavelength, the reflection suppression in sub-wavelength grating has been demonstrated in Figs. 6 and 7. It has also been shown that by overlapping of the optical modes, it is possible to cancel out the transmission or reflection. I disagree with the authors' claim that their structure illustrates "new optical resonance arising from the radiative coupling between arrayed silicon nanowires".
- 2) Color sensitive photodetector has been reported before using plasmonic gratings (Zheng, B. Y., Wang, Y., Nordlander, P., & Halas, N. J. Color-selective and CMOS-compatible photodetection based on aluminum plasmonics. *Advanced materials*, 26(36), 6318-6323 (2014)). Despite some differences, the idea of using resonances in a grating structure to attain color sensitivity is very similar to this work.
- 3) Reducing the undesired scattering or achieving the desired scattering in nanowire photodetectors have been reported by the same group previously (ref. [7] and [9]).

Besides, I have the following technical concerns:

- 4) In Fig. 4e, the authors have compared the reflection images in a very narrowband range and for one polarization, and they concluded that a conventional detector is not transparent. In my opinion this comparison is not fair. I think the comparison should be done for the entire spectrum and for both polarizations similar to what has been done in Fig. 2c. I assume that at least for the TE polarization there is no difference. Hence, under unpolarized illumination, even a nonoptimized photodetector should be transparent at some level. Also, the directional scattering in the optimized structure is narrowband. The question is by doing the optimization, how much improvement in the overall transmittance of three nanowire array is achievable?
- 5) The directional scattering is shown for normal incidences. For applications like what has been shown in Fig. 1.d, transparency of inclined incidences is important as well. It would be useful if the authors add one plot for another incident angle.
- 6) Page 2, lines 67-69: the sentence repeats the previous sentence without adding any new information.
- 7) Page 7, line 224: it might be more accurate if you say IQE is wavelength-independent in the visible range.

Reviewer #3:

Remarks to the Author:

The contents described in this paper seem to be the same level as the papers published in *Optics express* and *Optics Lett.* As an impression of the reviewer specializing in the optical communication device field, the structure as described in this paper has already been used in conventional vertical grating couplers and high-contrast grating couplers, and there is almost no difference in principle (Of course there may be a slight difference in theoretical approaches, but in the end it is the same idea). As the authors mention in this paper, it is true that there is a lot of previous researches related to filters (i.e., transmission characteristics control) that make use of the resonance of dielectric periodic nanostructure. But including them, the reviewer can not find clear novelty in this type of research.

From such a point of view, there is nothing new in the data shown in Fig. 2 where the transmission and reflection characteristics were observed by changing the structure of the Si grating. It seems better just to mention in Supplementary Information. It is even more important to actually

perform imaging by using the proposed technology as color pixels. And what the scale and sensitivity of imaging can be realized may help accept for publication in Nature Communications.

In that sense, the data shown in Fig. 3 is insufficient, and the results of some large-scale imaging experiments are strongly desired (Conversely, if the authors can show additional data, there is a possibility of acceptance for publication in Nature Communications). Also, although most experimental data are discussed using external quantum efficiency (EQE), from a practical point of view, the important factor is the measured value of photocurrent with respect to the light irradiation intensity per unit area. The authors should describe those data in addition to the EQE.

The above is the opinion of the reviewer who is engaged in research and development of practical optical communication devices. Therefore, from the viewpoint of researchers who are engaged in basic optics such as nanophotonics, the content of this paper may be very useful. Although my thinking is negative, I will leave the editors to make the final decision for Nature Communications.

Response to Reviewer 1:

Thank you very much for your very positive comments. A detailed response to each comment and/or question is given below.

1. *Analysis in Fig.1i would benefit from plotting both forward and backward scattering efficiencies for the nanowires. Also, the description states that this is the scattering from the nanowire array (rather than single nanowires). Could the authors specify how is the scattering from the arrays defined? I would expect the infinite arrays to be described in terms of transmission/reflection and only single nanowires or finite structures in terms of scattering.*

We agree that plotting both forward and backward scattering efficiencies in Fig. 1i would better illustrate the directional scattering from the designed NW array. We appreciate the suggestion and **have added the corresponding data in the revised Fig. 1i**. A discussion of the revised figure has been added to the manuscript as well.

The definition of the scattering cross section of a nanowire array used is the ratio of the integrated scattering power flow of each unit cell (period) and the incident plane wave intensity $\sigma_{sca} = \oint P_{sca} dl / P_{inc}$. The definition of the scattering efficiency is further normalized by the physical cross-section (period) of each NW $\sigma_{sca,eff} = \sigma_{sca} / p$.

As the reviewer argues, all the non-normal scattering plane waves should be canceled for a subwavelength infinite array. As such, the only possible non-zero plane-wave components must be along the normal direction. Thus, the reflection from an array is exactly the backward scattered plane wave and the transmission of an array is simply the superposition of the incident plane wave and the forward scattered plane wave. We use the term “scattering” for an array to be consistent with the definition for an individual NW, making quantitative comparison more straightforward. In this way, the analysis is always focused on the scattered field rather than switching to the full field description when we study a NW array.

2. *The authors write that “single NWs intrinsically do not support degenerate even and odd resonances for any cross-sectional geometry (Supplementary Fig. 1)”. This is quite important statement that motivates the authors to use the coupling between the nanowires in the arrays as an additional degree of freedom. However, the analysis shown in Supplementary Fig. 1 is not so straightforward to understand. To make it more clear I suggest the authors to plot spectral positions of different modes of a single nanowire for different aspect ratios (both below and above unity, at the moment only >1 aspect ratios are considered). This would simplify this analysis and clearly show whether or not the relevant modes can be spectrally overlapped.*

We appreciate this valuable suggestion. This statement is critical for our manuscript as it makes it very different from studies on directional scattering of single nanowires. As such, we agree that is essential that it is supported with a comprehensive modal analysis. **We have added an additional panel to Supplementary Fig. 1 (Supplementary Fig. 1c) to show the spectral positions of different modes of a single nanowire for different**

aspect ratios, per your suggestion. This figure clearly confirms the lack of degeneracy, even for aspect ratios below 1.

3. *The microscope images of the arrays shown in the insets in Fig.2a&b do not seem to be consistent with the spectra. The array is completely not seen in the transmission image for TE polarization, while the actual transmission is far from unity. At the same time, the same array is seen in the reflection image, while its actual reflection is very close to that of the substrate.*

We apologize for any confusion. We emphasize that the microscope images shown in Fig. 2a&b are filtered by a green color filter (center wavelength 550 nm, FWHM = 32 nm) to highlight the NW array behavior around the degenerate resonant wavelength. The wavelength range we used here is highlighted by a semi-transparent green band in Fig. 2a&b.

Within this range, the transmission for TE polarization (Fig. 2b, dashed dark green line) is almost identical to the transmission of a bare sapphire substrate (solid back line). This renders the right part of the microscope image (TE) almost invisible. A large difference in transmittance is observed for TM polarization however, which results in a strong contrast in the microscope image (left, TM).

In the meantime, although the reflection for TE/TM polarization (Fig. 2a) is only slightly lower than that of the surrounding sapphire substrate, the contrast settings used for the microscope image give rise to a difference that is large enough to be visible. Please note that the dynamic range of the microscope images in Fig. 2a&b are different. Fig. 2a compares very weak signals (low reflectance for both substrate and array), while Fig. 2b compares large signals (high transmittance for both substrate and array). As such, a small difference in intensity may give rise to large contrast in 2a, yet simultaneously small contrast in 2b.

4. *In the caption to Supplementary Fig.7, first (d) should be changed to (b).*

We have corrected this typo in the revised manuscript.

Response to Reviewer 2:

Thank you very much for your constructive and very useful comments. A detailed response to each comment is given below.

1. *The main claim of the paper is the suppression of the reflection using silicon nanowires array. The reflection of a silicon nanowire array has thoroughly been studied previously (e.g. Connie J. Chang-Hasnain and Weijian Yang, "High-contrast gratings for integrated optoelectronics," Adv. Opt. Photon. 4, 379-440 (2012)). Although the previous works have mostly focused on achieving strong reflection using high contrast grating with period around the wavelength, the reflection suppression in sub-wavelength grating has been demonstrated in Figs. 6 and 7. It has also been shown that by overlapping of the optical modes, it is possible to cancel out the transmission or reflection. I disagree with*

the authors' claim that their structure illustrates "new optical resonance arising from the radiative coupling between arrayed silicon nanowires".

We completely agree with the reviewer that the reflection and transmission of a silicon NW array has already been studied comprehensively using the method of Bloch mode analysis (e.g., ref [39] *Lalanne, P., Hugonin, J. P., Chavel, P. Optical properties of deep lamellar gratings: a coupled Bloch-mode insight. J. Light. Technol. 24, 2442-2449 (2006)*), as well as *Connie J. Chang-Hasnain and Weijian Yang, "High-contrast gratings for integrated optoelectronics," Adv. Opt. Photon. 4, 379-440 (2012)*). Briefly, in high contrast gratings different Bloch modes with different phase velocity can accumulate different phase retardation during propagation and interfere with each other to achieve either high reflection or high transmission around the designed wavelength.

However, the typical thickness of such high-contrast gratings is close to the designed wavelength. This can be understood intuitively as a certain propagation length is required to accumulate sufficient phase difference between the two Bloch modes. For a NW thickness much smaller than the wavelength we are in a very different operating regime. While the Bloch mode analysis may still work mathematically, it no longer provides useful physical insights anymore. Based on the different physics at work, it is more insightful to work with a different basis set as discussed below.

It is well established that in these deep-subwavelength NWs, the light-matter interaction can be understood and analyzed in terms of the excitation of different eigenmodes (optical resonances) confined in the individual NWs (ref [2,5], and *Fan, P. et al., "Optical Fano resonance of an individual semiconductor nanostructure" Nature Materials 13, pages 471–475 (2014)*). These eigenmodes have various radiation patterns with different symmetry and the interference between them results in the suppression or enhancement of scattering in designed directions/wavelengths. In this manuscript, we additionally find that the radiative coupling between neighboring silicon NWs gives rise to a new symmetric and highly tunable mode (middle inset in Fig. 1h). This new eigenmode is critical to achieve a degenerate optical resonance in NW arrays (Supplementary Fig. 1). **We emphasize that this mode is not the Bloch mode studied and used in high contrast gating papers**, as this mode is also confined in the vertical direction.

In conclusion, the Bloch mode analysis and optical resonance analysis are two different methods to study the optical properties of grating structures. Each has advantages and disadvantages. For the structure proposed in the manuscript, we argue that the optical resonance analysis is the better one for two reasons. First, the optical resonance analysis has a simpler physical picture as we avoid considering the complex reflections at the interface. Second, the optical resonance analysis naturally gives rise to the spectral and polarization sensitivity of NW arrays, which is at the basis of the design of a transparent spectro-polarimetric photodetector.

Finally, we want to emphasize that the main purpose of this manuscript is not merely to design a transparent NW array. Instead, we demonstrate the first deep sub-wavelength

thickness, transparent photodetection platform with intensity, color, and polarization sensing ability.

2. *Color sensitive photodetector has been reported before using plasmonic gratings (Zheng, B. Y., Wang, Y., Nordlander, P., & Halas, N. J. Color-selective and CMOS-compatible photodetection based on aluminum plasmonics. Advanced materials, 26(36), 6318-6323 (2014)). Despite some differences, the idea of using resonances in a grating structure to attain color sensitivity is very similar to this work.*

We appreciate the reviewer highlighting this important work that is related to our manuscript. **We have added this reference to the manuscript.** We agree that we share the concept of using optical resonances in a periodic structure for color sensing with this reference. However, the fundamental difference here is that by using the plasmonic grating structure, the strong back scattering from metallic grooves and the opaque substrate impede the implementation in transparent devices. Hence, the novelty of our work does not merely lie in color/polarization sensing using nanophotonic devices or realizing a transparent nanophotonic device. **It is the combination of transparency and nanophotonic spectro-polarimetric photodetection that makes this work distinguishable and novel,** as the optical resonances used in most sensing geometries naturally gives rise to strong back scatterings.

3. *Reducing the undesired scattering or achieving the desired scattering in nanowire photodetectors have been reported by the same group previously (ref. [7] and [9]).*

The reviewer is correct that our group focusses on the study of controlling light scattering/absorption in NWs for various optoelectronic applications. However, this manuscript is very different than the published papers (ref. [7] and [9]) for several reasons.

First, the current manuscript is the first time we demonstrate that the radiative coupling in arrayed NWs can lead to a degenerate optical resonance. In contrast, neither ref [7] nor ref [9] studied the optical modes in an infinite NW array.

Second, while both the photodetector proposed in the submitted manuscript and the one in ref [7] suppress the backscattering from NWs, a very different physical mechanism lies at the basis of the suppression. While ref [7] combines plasmonic and Mie resonances to cancel the backscattering, our current work employs semiconductor nanowire arrays to achieve for the first time a large-scale genuine transparent device configuration (i.e., the device is fabricated on a transparent substrate), which is critical to *in-situ* signal monitoring and VR/AR applications. Ref [7] focused on nanowires on an opaque gold substrate.

Third, ref [9] realized angle sensing for a specific incident wavelength and polarization. In the current work we focus on the *in-situ* multi-band color and polarization sensing for more practical applications.

4. In Fig. 4e, the authors have compared the reflection images in a very narrowband range and for one polarization, and they concluded that a conventional detector is not transparent. In my opinion this comparison is not fair. I think the comparison should be done for the entire spectrum and for both polarizations similar to what has been done in Fig. 2c. I assume that at least for the TE polarization there is no difference. Hence, under unpolarized illumination, even a nonoptimized photodetector should be transparent at some level. Also, the directional scattering in the optimized structure is narrowband. The question is by doing the optimization, how much improvement in the overall transmittance of three nanowire array is achievable?

We appreciate reviewer for raising this concern. It is correct that the comparison between the “optimized” and “conventional” photodetector is done for a narrowband range and for one polarization only. These results therefore cannot be extended directly to the situation of unpolarized white-light illumination. **We have rephrased the corresponding sentence in the manuscript to highlight the image taken condition.** However, we argue that it still makes sense to have such a comparison for several reasons. First, the main purpose of Fig. 1e is to highlight the huge change in optical response upon the realization of a degenerate optical resonance in a NW array. Thus, it would be better to focus on the wavelength range and the polarization supporting this degenerate optical resonance. Second, although the band we consider here is relatively narrow and for one polarization, it is already good enough to guarantee a good *in-situ* monitoring platform under polarized laser or even LED illumination, which covers a wide range of potential applications in optical communications. Third, although the “conventional” NW array is transparent for TE polarization, its overall averaged transmittance (<50%) is still considerably lower than the “optimized NW array” and thus cannot be used as a transparent detection system.

A quantitative analysis of the improvement is given in Supplementary Note 13. The bandwidth of the anti-reflection effect is fundamentally limited by the different quality factors of symmetric and anti-symmetric optical resonance in a NW array. One of the solutions to broaden the bandwidth of anti-reflection effect would be to multiplex two different types of NWs together (big-small-big-small). As a result, they would show second-order optical resonances at different wavelengths, effectively broadening the bandwidth of this resonant mode. Supplementary Fig. 13 shows the simulated reflection of a multiplexed NW array as a function of the incident wavelength and difference in width of two types of multiplexing NWs. A double-valley reflection spectrum is observed for multiplexed NW arrays. This is attributed to the fact that the “big” and “small” NWs show the second-order optical resonance at different wavelengths. Considering that the first-order optical resonance is broadband, the multiplexed NW array achieves the degenerate optical resonances at two different wavelengths, giving rise to the double-valley feature. Quantitatively, a 5%-10% overall transmittance improvement is expected as a result. In conclusion, we believe that further optimizations can lead the overall averaged transmittance approaching 80%, making the concept more promising for practical applications with unpolarized white-light illumination.

5. *The directional scattering is shown for normal incidences. For applications like what has been shown in Fig. 1.d, transparency of inclined incidences is important as well. It would be useful if the authors add one plot for another incident angle.*

We agree that the transparency of inclined incidences is very important for practical applications and are grateful for the suggestion. All the optical spectra in the manuscript were taken with a 20X (NA=0.4) objective, indicating that the inclined incidences up to 23° do not significantly change the optical response noticeably. To corroborate this, **we added simulated reflection and transmission spectra for three different NW arrays under inclined illumination (10°) as shown in Supplementary Fig. 5c&d.**

6. *Page 2, lines 67-69: the sentence repeats the previous sentence without adding any new information.*

Thank you very much for highlighting this. At the beginning of this paragraph before this sentence (*Page 2, lines 57-67*), we specify the key difference between having degenerate optical resonance and the conventional Kerker condition. Then, this sentence and the following sentence further claim that although the degenerate optical resonance can be realized in individual Si nanodisks (*Page 2, lines 67-69*), we cannot find the same degenerate mode in single NWs (*Page 2, lines 69-70*). **We have rephrased this sentence to make it clearer that Si nanodisks are one example but individual NWs are a different story.**

7. *Page 7, line 224: it might be more accurate if you say IQE is wavelength-independent in the visible range.*

Thank you very much for your suggestion. **We have corrected this in the revised manuscript.**

Response to Reviewer 3:

We appreciate the reviewer's valuable comments from the viewpoint of a practical optical communications expert. The corresponding responses are as listed below.

The contents described in this paper seem to be the same level as the papers published in Optics express and Optics Lett. As an impression of the reviewer specializing in the optical communication device field, the structure as described in this paper has already been used in

conventional vertical grating couplers and high-contrast grating couplers, and there is almost no difference in principle (Of course there may be a slight difference in theoretical approaches, but in the end it is the same idea). As the authors mention in this paper, it is true that there is a lot of previous researches related to filters (i.e., transmission characteristics control) that make use of the resonance of dielectric periodic nanostructure. But including them, the reviewer cannot find clear novelty in this type of research.

We fully agree with the reviewer that grating structures have been widely used for color sorting functions. However, the fundamental difference between this work and the others is that we combine the color/polarization sorting and detection functionalities in a carefully designed transparent NW array. This is far from trivial as the optical resonances commonly used for color sorting intrinsically give rise to strong back scattering. As a result, the overall transmittance of conventional nanophotonic grating color filters (ref [16-21]) is only about 30% even before the photodetection process. Thus, it's impossible to design a real transparent photodetection platform by using these conventional nanophotonic grating/dye-based color filters. To the best of our knowledge, the technology demonstrated in this manuscript is the first ultra-compact (deep subwavelength thickness) transparent spectro-polarimetric detection system.

From such a point of view, there is nothing new in the data shown in Fig. 2 where the transmission and reflection characteristics were observed by changing the structure of the Si grating. It seems better just to mention in Supplementary Information. It is even more important to actually perform imaging by using the proposed technology as color pixels. And what the scale and sensitivity of imaging can be realized may help accept for publication in Nature Communications.

We are sorry not to emphasize the novelty of the data in Fig. 2 clear enough. Fig. 2a & 2b show the perfect suppression of reflection and enhancement of transmission when the NW arrays are on resonance, which is opposite to the situation in conventional resonant grating structures. Optical resonances usually result in the enhancement of backscattering/reflection, e.g., see red lines in Supplementary Fig. 7a-d. We emphasize that the results presented in Fig. 2 are not simply diffraction effects. In fact, the inter-wire spacing is too small to enable free-space diffraction for all geometries shown in the manuscript. Besides the spectrally engineered resonant coatings, we further show that the overall averaged transmittance reaches 70%, both in simulations and experiment (in contrast, conventional grating color filter only has ~30% overall transmittance). Finally, the reflection optical images visualize the transparency clearly in the whole visible spectrum range. In conclusion, we argue that the data shown in Fig. 2 is new and important to support our statement that we demonstrate a transparent spectro-polarimetric detection system based on localized resonant eigenmodes in this manuscript.

The focus of this work is the first demonstration and study of a resonant transparent spectro-polarimetric detection system. It is focused on highlighting the essential physics and we believe that creating densely integrated pixel array for imaging is an engineering task of a very different nature and beyond the scope of this manuscript.

In that sense, the data shown in Fig. 3 is insufficient, and the results of some large-scale imaging experiments are strongly desired (Conversely, if the authors can show additional data, there is a

possibility of acceptance for publication in Nature Communications). Also, although most experimental data are discussed using external quantum efficiency (EQE), from a practical point of view, the important factor is the measured value of photocurrent with respect to the light irradiation intensity per unit area. The authors should describe those data in addition to the EQE.

We fully agree that further experiments on large-scale imaging processing are exciting routes forward to realize the commercialization of the proposed concept here. However, the focus of the current work is on the fundamental understanding and control of light-matter interaction at the nanoscale. We aim to generate theoretical and intuitive understanding of the realization of degenerate optical resonances, and how to leverage this to achieve a transparent resonant NW array. Moreover, we stress that this first demonstration of transparent arrays has practical applications as invisible, color and polarization sensitive photodetectors. The continuous pursuit of technological implementation is a logical next step – yet a topic of a future manuscript.

In the experiment, we use a 50x (NA=0.42) objective to focus the light into a $\sim 1.5 \mu\text{m}$ spot on the NW array photodetector. The measured responsivity is $\sim 5 \times 10^{-6} \text{A/W}$ when the NW array is on resonance. As we mentioned in the main text, in this study we have not aimed to optimize the electrical contacts or the NW surface passivation to achieve high efficiency. Despite the low EQE/responsivity, we emphasize that the spectral shape of the EQE spectrum and the uniform photocurrent generation provide an irrevocable proof of the proposed concept. These measurements demonstrate reliable color, polarization, and intensity detection in a transparent ultra-thin photodetector geometry. Mature processing techniques, including standard passivation techniques, a p-n junction, and local heavy doping at the contacts can improve the EQE of NW photodetectors for practical applications.

The above is the opinion of the reviewer who is engaged in research and development of practical optical communication devices. Therefore, from the viewpoint of researchers who are engaged in basic optics such as nanophotonics, the content of this paper may be very useful. Although my thinking is negative, I will leave the editors to make the final decision for Nature Communications.

We would like to reiterate our gratitude for your valuable comments from a practical optical communication device expert perspective. As we mentioned above, we believe that the main strengths of our results lie in the broad interest in the field of nanophotonics. We also believe that the further development of this concept (i.e., large-scale image processing and improvement of EQE) will lead to practical products in the future.

Reviewers' Comments:

Reviewer #1:

Remarks to the Author:

The authors have adequately addressed all my comments and further improved their manuscript. It can now be accepted for publication in Nature Communications.

Reviewer #2:

Remarks to the Author:

I appreciate the authors' effort to address my comments, but I am afraid that I am not convinced this work should be published in Nature Communications.

From the fundamental point of view, Bloch mode analysis and optical resonance analysis reported here are just two different modal bases to explain the same problem. I agree with the authors that the resonance analysis gives a simpler picture for the reflection cancellation. However, it does not look to be an important ingredient for the novelty. As I mentioned in my initial report, the reflection cancellation because of the destructive interference between the Bloch modes have already been demonstrated in high contrast gratings even in sub-wavelength regime.

From the practical point of view, I am not convinced that the overall performance of the "optimized" photodetector (Fig. 1g) is considerably improved compare to the "conventional" one (Fig. 1f). I think it was a reasonable request to compare the overall averaged transmission for these two structures in the way illustrated in Fig. 2. Also, a plot of the external quantum efficiency for the conventional photodetector is needed to see if there is any trade-off between EQE and transparency. Moreover, it seems that the reflection cancellation is mostly optimized for normal incidences, which significantly limits their applications (it would have been more helpful if the authors had reported the transmission spectrum for incident angles greater than 10 degree).

Reviewer #3:

Remarks to the Author:

From the viewpoint of researchers who are engaged in basic optics, the content of this paper will be very useful.

Response Letter

Transparent Multispectral Photodetectors Mimicking the Human Visual System

Qitong Li¹, Jorik van de Groep¹, Yifei Wang¹, Pieter G. Kik^{1,2}, Mark L. Brongersma^{1*}

1. Geballe Laboratory for Advanced Materials, Stanford University, Stanford, USA

2. CREOL, The College of Optics and Photonics, University of Central Florida, Orlando, USA

* Email: brongersma@stanford.edu

List of changes to the manuscript and supporting information:

1. We added an additional Supplementary Note (Supplementary Note 4 in the revised version) to give a comprehensive comparison between “conventional” NW detectors and “transparent” NW detectors.
2. We added the simulated reflection and transmission for three our structures as a function of incident wavelength and angle of incidence, as shown in Supplementary Fig. 6e-j.
3. We added one sentence at the end of the section *Description of the transparent resonant NW arrays* in the main text to introduce the comprehensive comparison we make in Supplementary Note 4.

Detailed response to the comments from the reviewers:

Reviewer #1 (Remarks to the Author):

The authors have adequately addressed all my comments and further improved their manuscript. It can now be accepted for publication in Nature Communications.

Response to Reviewer 1:

We appreciate the reviewer’s constructive comments which helped us improve the manuscript quality significantly.

Reviewer #2 (Remarks to the Author):

I appreciate the authors' effort to address my comments, but I am afraid that I am not convinced this work should be published in Nature Communications.

From the fundamental point of view, Bloch mode analysis and optical resonance analysis reported here are just two different modal bases to explain the same problem. I agree with the authors that the resonance analysis gives a simpler picture for the reflection cancellation. However, it does not look to be an important ingredient for the novelty. As I mentioned in my initial report, the reflection cancellation because of the destructive interference between the Bloch modes have already been demonstrated in high contrast gratings even in sub-wavelength regime.

From the practical point of view, I am not convinced that the overall performance of the "optimized" photodetector (Fig. 1g) is considerably improved compare to the "conventional" one (Fig. 1f). I think it was a reasonable request to compare the overall averaged transmission for these two structures in the way illustrated in Fig. 2. Also, a plot of the external quantum efficiency for the conventional photodetector is needed to see if there is any trade-off between EQE and transparency. Moreover, it seems that the reflection cancellation is mostly optimized for normal incidences, which significantly limits their applications (it would have been more helpful if the authors had reported the transmission spectrum for incident angles greater than 10 degree).

Response to Reviewer 2:

We thank the reviewer again for all the comments that aim to improve the manuscript quality. A detailed response to each comment is given below.

- 1. From the fundamental point of view, Bloch mode analysis and optical resonance analysis reported here are just two different modal bases to explain the same problem. I agree with the authors that the resonance analysis gives a simpler picture for the reflection cancellation. However, it does not look to be an important ingredient for the novelty. As I mentioned in my initial report, the reflection cancellation because of the destructive interference between the Bloch modes have already been demonstrated in high contrast gratings even in sub-wavelength regime.*

We apologize for any potential confusion. However, we re-emphasize that the purpose of this manuscript is not to merely design a subwavelength-thick grating structure with high transmittance. **It is the combination of transparency and nanophotonic spectro-polarimetric photodetection that makes this work distinguishable and novel**, as the optical resonances used in most sensing geometries naturally give rise to strong back scatterings.

2. *From the practical point of view, I am not convinced that the overall performance of the "optimized" photodetector (Fig. 1g) is considerably improved compare to the "conventional" one (Fig. 1f). I think it was a reasonable request to compare the overall averaged transmission for these two structures in the way illustrated in Fig. 2.*

We appreciate the reviewer for raising this reasonable request. **We added Supplementary Note 4 to address the reviewer's concern.** In Fig. 1e we used a narrow band illumination to highlight the contrast between "conventional" and "transparent" NW arrays. However, we emphasize that even under the white light illumination, the "transparent" NW array still shows strongly enhanced transparency under TM polarization, as shown in Supplementary Fig. 4a-c. The averaged transmittance of the "conventional" NW array is only $\sim 30\%$.

Overall, unlike electromagnetic induced transparency (EIT), the optimized array does not sacrifice broadband transparency to achieve high transparency at the target wavelength. The non-perfect anti-reflection effect away from the resonance is due to the difference in quality factor between the symmetric and anti-symmetric modes. However, for the "conventional" array with separated resonant modes, the reflection is significantly higher as no modal interference mechanism can be used to cancel the strong back scattering from the individual resonant modes.

3. *Also, a plot of the external quantum efficiency for the conventional photodetector is needed to see if there is any trade-off between EQE and transparency.*

We appreciate this valuable suggestion, and have implemented this in the new Supplementary Note 4. **We emphasize that there is no trade-off between EQE and transparency.** On the contrary, the EQE (absorption) is further enhanced by inducing the degenerate optical resonant modes in the "transparent" NW arrays.

To demonstrate this, we use the reasonable assumption that with the same fabrication process, the "conventional" NW detector should have an IQE similar to that of the "transparent" NW detector. As a result, the comparison of absorption spectra is sufficient to compare the optoelectronic performance of the two structures. Supplementary Fig. 4d clearly shows that the degenerate optical modes in the "transparent" detector enhance the absorption by 4 times compared to the "conventional" array, and the resulting sharper absorption peak is also critical in the color detection process. As such, the EQE of the "transparent" detector outperforms that of the "conventional" detector.

4. *Moreover, it seems that the reflection cancellation is mostly optimized for normal incidences, which significantly limits their applications (it would have been more helpful if the authors had reported the transmission spectrum for incident angles greater than 10 degree).*

We appreciate this suggestion. **We added more detailed reflection and transmission data as a function of incident wavelength and angle of incidence (0°-20°) to supplementary Fig. 7.** We agree that the reflection cancellation is optimized near normal incidence, and the transparency degrades with increasing angle of incidence because of both the first-order diffraction and the excitation of a guided mode resonance in the “red” NW arrays. However, we argue that the transparency performance is still acceptable at 10° as shown in supplementary Fig. 7c-d. **Keeping in mind that the central field of view of human vision system is only ~ 13°** (Younis, O., Al-Nuaimy, W., Alomari, M. H., & Rowe, F. (2019). *A Hazard Detection and Tracking System for People with Peripheral Vision Loss using Smart Glasses and Augmented Reality. Int. J. Adv. Comput. Sci. Appl, 10, 1-9*), the demonstrated angular tolerance should be sufficient for a wide range of potential applications involving augmented reality technologies and free-space optical communications.

Reviewer #3 (Remarks to the Author):

From the viewpoint of researchers who are engaged in basic optics, the content of this paper will be very useful.

Response to Reviewer 3:

We gratefully thank the reviewer for the positive conclusion and all the constructive comments from the point of view of an expert in the field of practical optical communication.

Reviewers' Comments:

Reviewer #2:

Remarks to the Author:

The revised version of the manuscript represents some improvement, and the authors have addressed most of my concerns. Now, I can recommend publication in Nature Communications.

Response Letter

Transparent Multispectral Photodetectors Mimicking the Human Visual System

Qitong Li¹, Jorik van de Groep¹, Yifei Wang¹, Pieter G. Kik^{1,2}, Mark L. Brongersma^{1*}

1. *Geballe Laboratory for Advanced Materials, Stanford University, Stanford, USA*

2. *CREOL, The College of Optics and Photonics, University of Central Florida, Orlando, USA*

* *Email: brongersma@stanford.edu*

Reviewer #2 (Remarks to the author):

The revised version of the manuscript represents some improvement, and the authors have addressed most of my concerns. Now, I can recommend publication in Nature Communications.

Response to Reviewer 2:

We really appreciate it for reviewer's positive opinion and all the valuable comments made that improved the manuscript quality a significantly.